# Continuance Intention to Use and Perceived Net Benefits as Perceived by Streaming Platform Users: An Application of the Updated IS Success Model

**DOI:** 10.3390/bs12050124

**Published:** 2022-04-24

**Authors:** Chan-Sheng Kuo, Chia-Chien Hsu

**Affiliations:** 1Department of Information Management, Shih Hsin University, Taipei City 116, Taiwan; cskuo@mail.shu.edu.tw; 2Department of Tourism, Shih Hsin University, Taipei City 116, Taiwan

**Keywords:** the updated IS Success Model, streaming platform, user satisfaction, perceived net benefit, continuance intention to use

## Abstract

Video game streaming has become a popular leisure activity which, depending on content, level of skill, and user interest, reaches a wide array of audiences. Using streaming platforms, client users are able to broadcast their video gameplays and use chat rooms to interact with other viewers and game players in real time. This paper aimed to develop and examine a theoretical explanation concerning the formation of client users’ satisfaction with, perceived net benefits of, and continuance intention to use a particular streaming platform. The study employed the updated IS Success Model. An online questionnaire was designed for individuals who were game streaming users of a streaming platform. A total of 632 usable responses were used in the data analysis. The results pointed out that system quality, information quality, service quality, and user motivation all positively and significantly affected user satisfaction. User satisfaction had a positive and significant effect on perceived net benefits and continuance intention to use a streaming platform. Perceived net benefits positively and significantly related to continuance intention to use a particular streaming platform. The moderating effect of emotional involvement between user satisfaction and perceived net benefits was confirmed.

## 1. Introduction

Video game streaming has become a popular leisure activity which, depending on content, level of skill, and user interest, reaches a wide array of audiences. Using streaming platforms, client users are able to broadcast their video gameplays and use chat rooms to interact with other viewers and game players in real time. A streaming platform, such as Twitch and YouTube Gaming, is an on-demand online source that provides access to a wide variety of entertainment venues for client users [1]. These platforms enable users to obtain access to movies, tutorial programs, competitive gameplays, or online chat rooms so that users can customize their experience based on their individual preferences [2].

As the technology becomes more user-friendly and easier to access for those who have access to a computer or the newest gaming consoles, the members of the live-streaming community have rapidly increased [3]. Basically, game streaming users can be categorized into two groups: streamers and viewers [4]. Streamers generate content or information while viewers receive or consume the content or information produced by streamers [4]. More specifically, streamers engage in game knowledge and skill performance via live streaming and subsequently viewers are able to watch and learn in real time. Therefore, the capacities and characteristics of video game streaming platforms used by streamers and viewers is important to discover in order to better understand the dynamics of this technology.

Streaming platforms need to make continuous improvements in order to meet the often dynamic and changing needs of users. Additionally, if the capacity of a platform is responsive and reactive to user satisfaction, the platform will be likely to increase its user base. Researchers [4] indicate that “platform impact” plays a pivotal role in the behavior of game streaming users. Therefore, a better understanding of video streaming platforms’ information systems as perceived by users is a focal point not only to researchers, but also to those platform operators who are actually making efforts to know their client users’ behaviors.

In the literature, many frameworks have been proposed to assess the success of information systems. One of the most validated measures is the Information Systems Success Model (IS Success Model) developed by DeLone and McLean [5,6]. Based upon the communication research and management information research (MIS), they proposed an interrelated link between use/user satisfaction and organizational impact mediated by individual impact. Both use and user satisfaction are shaped by system quality and information quality in the 1992 model they developed. In 2003, DeLone and McLean proposed an updated model. Service quality was included as one of the variables in terms of user satisfaction and use/intention to use. Intention to use was added as an alternative variable because “simply saying that more use will yield more benefits, without considering the nature of this use, is clearly insufficient.” [6] (p. 16). DeLone and McLean grouped two variables (i.e., individual impact and organizational impact) into one (i.e., net benefits) in order to be more inclusive in the future application of the IS Success Model. In this study, the framework of the IS Success Model is used in order to test the proposed hypotheses which guide this investigation.

Video game streaming has made huge strides in adoption and use [7]. Client users across the world can select their favorite entertainment programs as well as engage in social interactions using streaming platforms. Thus, the purpose of this study was to examine streaming platform users’ satisfaction and how the level of satisfaction may lead to users’ benefits and intentions to use a particular streaming platform in a continuous and dedicated fashion. However, it was discovered that little research exists on streaming platforms’ capacities to nurture their connections with client users. Accordingly, no study has applied the updated IS Success Model in testing client users’ intentions to continue to use a chosen streaming platform. This study enables researchers to see the insights from the client-users’ perspective and can hopefully assist platform developers and operators to make improvements to streaming platforms and produce more user-friendly programs.

## 2. Literature Review and Hypothesis Development

### 2.1. The IS Success Model and Updated IS Success Model

Emphasizing IS effectiveness, DeLone and McLean [5,6] initially proposed the IS Success Model in 1992 and presented the updated model in 2003. In the original IS Success Model (Figure 1), DeLone and McLean identified six interrelated variables based upon a review of the research published from 1981 to 1987 [5]. These variables include system quality, information quality, use, user intention, individual impact, and organizational impact. After the IS Success Model was published, IS researchers started to call for model modifications, while the authors asked researchers for further development and validation [8]. Seddon and Kiew [9], for example, applied a part of the IS Success Model and believed that the variable, use, should be referred to the concept of perceived usefulness used in the Technology Acceptance Model developed by Davis [10].

Based on a series of reviews of empirical studies, DeLone and McLean [6] made certain modifications by consolidating existing variables and incorporating new variables into the original model (Figure 2). Following the suggestions provided by Pitt, Watson, and Kavan [12], the updated model includes ‘service quality’ as a new variable. Most importantly, the updated model clarifies the variable ‘use’. In the original model, a difficulty in interpreting the variable ‘use’ exists. DeLone and McLean [6] recommend the variable ‘intention to use’ as an alternative to address the rationale that ‘use’ must proceed ‘user satisfaction’ in the process. Accordingly, positive experiences will result in a higher intention to use and contribute to the presence of the purported measure (e.g., improved productivity and streaming platform revisit). Finally, individual impact and organizational impact in the original model are categorized into the variable ‘net benefits’. Such modifications enable the model to be more applicable in general [8].

### 2.2. User Satisfaction

Satisfaction is a relative term between individual expectations and actual situations [13]. An individual would be satisfied if actual situations meet his/her expectations, and vice versa. In the language of consumer behavior, “if the actual outcomes are judged to be less than what the buyer expected, the buyer feels dissatisfied” [14] (p. 139). Using this concept, a common understanding concerning user satisfaction would be that the level of satisfaction possessed by a user would impact future attractiveness of, and interactions with, a particular subject. In the field of IS, user satisfaction refers to the feelings and attitudes, gathered from a variety of perceived benefits, that an individual would like to receive from an IS interaction [11,15]. In the IS Success Model literature, user satisfaction is defined as “users’ level of satisfaction with reports, web sites, and support services” [8] (p. 239). That is, a user is likely to use a particular system in a continuous fashion because he/she receives positive experiences and is satisfied with the services provided by the IS.

### 2.3. Perceived Net Benefits

To make the updated IS Success Model more parsimonious, DeLone and McLean [6] combined individual impact and organizational impact into the construct ‘net benefits’. This construct is considered one of the most important measures in the model [16] and generally refers to the difference between positive and negative outcomes [6,17]. What net benefits need to be measured relies on the system or level of impacts being assessed [18]. Many researchers [19] focus on the value of the technology investment via quantifiable measures (e.g., cost, time, productivity, and market share), whereas others [20] argue that the measure of net benefits concerning numeric analyses can be a difficult task because the intangibles related to the system used, and other compounding environmental variables, may have an effect on such measures.

Applied to the field of IS, net benefits refer to the extent to which IS contributes to the successes of target subjects [8]. However, how to objectively measure net benefits is subject to interpretation. In fact, measuring IS users’ perceptions has become more pertinent in many cases [11,16]. As Gable, Sedera, and Chan [21] stated, IS success is a multidimensional phenomenon. DeLone and McLean [6] state that IS success directly infers that net benefits is a multidimensional construct and its measure needs to be determined by research context and goals of the investigation. In this study, perceived net benefits is applied to mean that a client user believes that the use of a streaming platform will result in positive outcomes.

### 2.4. Continuance Intention to Use

Though DeLone and McLean [6] argue that ’use’ is a proper measure in the original IS Success Model, the simplicity of the ’use’ definition is indeed unable to provide sufficient explanations in terms of the considerations of the extent, nature, quality, and appropriateness of the term ’use’. In the updated IS Success Model, DeLone and McLean [6] employ ‘intention to use’ as an alternative variable for the purpose of further explaining the context of the variable ‘use’. This is due to the debate concerning whether use can be regarded as a positive measure of IS success [22].

The continuance of intention to use is defined as the likelihood an individual retains or continues to use a particular application [11]. This construct broadens the meaning of ’use’ and enables DeLone and McLean to address certain degrees of the process of using versus causal concerns related to use revealed by Seddon [22]. In the literature, intention to use can be interpreted as a predictive variable similar to the variable attitude toward using in Davis’ Technology Acceptance Model. As DeLone and McLean [6] specified, “intention to use is an attitude, whereas use is a behavior” (p. 23). Researchers are able then, based on their research needs, to make use of both terms. In this study, continuance intention to use is employed as an attitude and a dependent variable for user satisfaction. This is because the respondents of the study were client users using a particular streaming platform when responding to the questionnaire.

### 2.5. User Satisfaction and Antecedent Variables

System quality refers to the desirable characteristics of a system [8]. It is one of the success dimensions presented in both the original and updated IS Success Model. This dimension mainly addresses the usability and performance functions of a system under investigation [18]. Perceived ease of use employed in the Technology Acceptance Model is a common measure in the literature related to system quality [10]. Other measures may consist of access, usability, navigation, interactivity, user friendliness, flexibility, convenience, accuracy, reliability, or timeliness [23,24,25]. In this study, the measures of system quality focus on access, user friendliness, and interactivity of a particular streaming platform. In the literature, a considerable body of studies [11,23,26,27] has shown that the relationship between system quality and user satisfaction is positive and as such, has led to the following hypothesis:

**Hypothesis** **1** **(H1).**
*A streaming platform’s system quality is positively associated with user satisfaction.*


Being presented in both the original and the updated IS Success Model, information quality is defined as the desirable characteristics associated with the system outputs [8]. Such characteristics may consist of relevance, accuracy, understandability, completeness, timeline, conciseness, importance, and usability [6,8,21]. In this study, the measures of information quality focus on accuracy, completeness, and understandability. Similar to system quality, a considerable body of research [11,23,28] has indicated that the relationship between information quality and user satisfaction is positive and as such, has led to the following hypothesis:

**Hypothesis** **2** **(H2).**
*A streaming platform’s information quality is positively associated with user satisfaction.*


Service quality refers to the quality of the supports that the users of a particular system receive from the service provider [8,18]. As an updated dimension of the IS Success Model, the measures of service quality may include responsiveness, interpersonal quality, reliability, assurance, tangibles, empathy, IS training, and flexibility [12,18,29]. In this study, the measures of this dimension consisted of responsiveness, reliability, and interpersonal quality. Prior research [30,31] has revealed that the relationship between service quality and user satisfaction is positive and as such, has led to the following hypothesis:

**Hypothesis** **3** **(H3).**
*A streaming platform’s service quality is positively associated with user satisfaction.*


In this study, the research team extended the model and employed user motivation as one of the antecedent variables in terms of user satisfaction. User motivation is defined as an intrinsic factor which directs and arouses individual behavior in using a particular streaming platform. This definition is derived from the approach of leisure studies which emphasizes intrinsic motivation because “leisure activities have no obvious external forces compelling individuals to engage in particular activities” [32] (p. 219). The characteristics of user motivation focus on the ideas of perceived freedom, self-determination, and the voluntary nature of participation [32,33]. That is, a user can be allowed to freely select and pursue a streaming platform and various programs (i.e., live gaming program and chat room communication) provided by the streaming platform. In this study, the leisure motivation scale (LMS) developed by Beard and Ragheb [32] was employed to measure a user’s motivation for engaging in leisure activities. The 12-itemed LMS includes four dimensions: intellectual, social, competence-mastery, and stimulus-avoidance. The measures of the study focus on social, competence-mastery, and stimulus-avoidance dimensions. Since the respondents of the study were streaming platform users, three statements from LMS were selected, modified, and employed for the purpose of addressing the characteristics of streaming platform users and the representation of each dimension holistically. Prior research [34,35] has revealed that the relationship between user motivation and satisfaction is positive and as such, has led to the following hypothesis:

**Hypothesis** **4** **(H4).**
*A streaming platform’s user motivation is positively associated with user satisfaction.*


### 2.6. User Satisfaction and Perceived Net Benefits

Specified in the updated IS Success Model, user satisfaction is the antecedent variable of net benefits and a widely examined construct in the IS field [18]. A number of instruments have been developed for the purpose of measuring user satisfaction such as the end-user computing satisfaction instrument [36,37] and the user information satisfaction instrument [15]. In this study, user satisfaction refers to game players’ level of satisfaction with the services provided by a particular streaming platform. The measures of the user satisfaction construct focus on adequacy, enjoyment, information satisfaction, and overall satisfaction.

As the overall dependent variable of the updated IS Success Model, perceived net benefits subsume the dimensions of individual and organizational impacts of the original IS Success Model [18]. The essence of net benefits centers on how client users perceive the advantages of using a particular streaming platform. Effectiveness, learning, productivity, usefulness, or task performance are the typical dimensions used to measure net benefits. In this study, the measures of net benefits focus on usefulness, learning, and task performance. Prior research [27,38,39] has revealed that the relationship between user satisfaction and perceived net benefits is positive and as such, has led to the following hypothesis:

**Hypothesis** **5** **(H5).**
*A streaming platform’s user satisfaction is positively associated with perceived net benefits.*


### 2.7. User Satisfaction and Continuance Intention to Use

As a newly proposed variable in the updated IS Success Model, continuance intention to use is defined as the likelihood that an individual employs a particular IS system or an application in the future [11]. Research on the IS literature has indicated that continuance intention to use can be mainly determined by user satisfaction with positive IS usage [40,41]. In the social media usage context, research has shown that user satisfaction is the primary determinant in terms of user’ continuance intention [42,43]. In the literature, a positive relationship between user satisfaction and continuance intention to use is well documented [41]. In this study, as an individual decision to use a streaming platform in a continuous fashion, the user is likely to be satisfied with the product/service provided by a particular service provider. Thus, the relationship between user satisfaction and continuance intention to use is positive and as such, has led to the following hypothesis:

**Hypothesis** **6** **(H6).**
*A streaming platform’s user satisfaction is positively associated with continuance intention to use.*


### 2.8. Perceived Net Benefits and Continuance Intention to Use

In this study, the research team extended the model and examined the relationship between perceived net benefits and continuance intention to use. Continuance intention is always considered as a dependent variable in the literature. Zheng et al. [44] examined the relationship between continuance intention and individual benefits as perceived by information-exchange virtual communities. Han et al. [45] investigated the relationship between customer perceived benefits and users’ continuance intention in online brand communities. The results of the above studies indicate that a positive relationship exists between perceived benefits and continuance intention. In this study, as a user perceives that using a streaming platform can be helpful in learning and task performance, he/she is likely to use that particular streaming platform in a continuous fashion. As such, the following hypothesis is proposed:

**Hypothesis** **7** **(H7).**
*A user’s perceived net benefits is positively associated with continuance intention to use a streaming platform service.*


### 2.9. Moderating Effects

Emotional involvement is the cornerstone of emotional experiences during media reception and refers to “the degree to which a media user is emotionally engaged with a media experience, content, or character and is experiencing intense feelings” [46] (p. 23). The research team believes that an emotionally involved person is likely to be “caught” by a media-rich environment. This is because the presences of fascinating pictures, affective music, and emotionalized scenes are all possible components which induce emotional involvement from client users [46]. In addition, Kowert et al. [47] indicate that online gaming programs enable multi-players to share socially accommodating environments. Through the gaming programs provided by streaming platforms, cyber-based social environments allow client users to engage in effective communications [48] and further enhance their online social relationships [49]. As a result, getting emotionally involved in online gaming has become a developmental process which enables players to receive benefits (i.e., friend-based supports) and to enhance their continuance intention to use a particular streaming platform.

The object under investigation is how emotional involvement affects the relationships among user satisfaction, perceived net benefits, and continuance intention to use a particular streaming platform. In general, the more users are satisfied and emotionally involved in a streaming platform’s programs, the more likely those users increase the likelihood to perceive benefits are gained and to have intention to use the platform in a continuous fashion. However, variation in emotional involvement may have an effect on proposed relationships. A user with lower emotional involvement may feel reluctant to spend long hours using streaming platform programs. In contrast, an emotionally involved user not only engages in his/her favorite programs, but also has a strong tendency to share individual experiences, knowledge, or intimate information with others [50]. That is, the relationships among user satisfaction, perceived net benefits, and continuance intention can be attenuated for users who are low in emotional involvement. No prior research exists to explain the relationships among these proposed variables. Therefore, the research team proposed that emotional involvement will affect the relationships among user satisfaction, perceived net benefits, and continuance intention to use a streaming platform and as such, has led to the following hypothesis:

**Hypothesis** **8** **(H8).**
*A streaming platform user’s emotional involvement moderates the relationship between user satisfaction and perceived net benefits.*


**Hypothesis** **9** **(H9).**
*A streaming platform user’s emotional involvement moderates the relationship between user satisfaction and continuance intention.*


This above section addresses whether user satisfaction is related to perceived net benefits and continuance intention to use, and whether user satisfaction is associated with the antecedent variables (i.e., system quality, information quality, service quality, and user motivation) and moderating variables (i.e., emotional involvement). A conceptual model was developed and is presented in Figure 3.

## 3. Methods

### 3.1. Measures

Based on existing theories and an extensive review of literature, a questionnaire was developed in order to test the hypotheses of this study. The questionnaire included two sections. Section I was to assess the constructs of this study including system quality, information quality, service quality, user motivation, emotional involvement, user satisfaction, perceived net benefits, and continuance intention to use. Section II consisted of statements that were collected for demographic information. Regarding the response categories of this study, a seven-point Likert-type scale ranging from 1 (strongly disagree) to 7 (strongly agree) was employed.

System quality was measured by four statements adopted from Petter et al. [8], Wu and Wang [11], Zheng et al. [44] (2013), and Prybutok et al. [51]. Information quality was measured by four statements adopted from Wu and Wang [11], Zheng et al. [44], and Prybutok et al. (2008). Service quality was measured by three statements adopted from Petter et al. [8], Zheng et al. [44], and Prybutok et al. [51]. User motivation was measured by three statements adopted from Beard and Ragheb [32]. Emotional involvement was measured by three statements adopted from Wirth et al. [46]. User satisfaction was measured by five statements adopted from Urbach and Muller [18] and Petter et al. [8]. Perceived net benefits were measured by six statements adopted from Han et al. [45] and Prybutok et al. [51]. Continuance intention was measured by four statements adopted from Petter et al. [8], Urbach and Muller [18], Han et al. [45], Fagan et al. [52], Chen et al. [53], and Bhattacherjee [54]. In sum, a total of 29 statements were included for measuring seven constructs.

The face and content validity of the survey instrument was assessed by a panel of experts (n = 5). The panel experts’ opinions were assessed, and modifications were made for the purpose of refining the survey instrument. A pilot test with 35 individuals who were client users of a particular streaming platform was conducted for the purpose of examining the feasibility of statements. The Cronbach’s α coefficients of system quality, information quality, service quality, user motivation, user satisfaction, perceived net benefits, and continuance intention were 0.86, 0.89, 0.79, 0.85, 0.86, 0.88, and 0.86, respectively. The instrument was considered both valid and reliable.

### 3.2. Data Collection and Respondents

Purposeful sampling was used in this study. This is because a complete list of the population was unavailable. An online questionnaire was designed for individuals who were game streaming users of a streaming platform. Game streaming users were first asked to participate in this study. A link was provided for the purpose of enabling participants to complete the designed questionnaire. The respondents of the study were at least 18 years old and were informed that their participation was voluntary and that their responses would be kept strictly confidential, data would only be used in aggregate, and no single data point would be used, related, or identified to a single respondent. The period of data collection lasted about three months from 6 April 2018 to 12 July 2018. Of the 705 respondents, 73 responses were unusable due to missing data and/or those respondents who did not have the experience of participating in live-streaming programs. As a result, a total of 632 usable responses were used in the data analysis.

## 4. Results

### 4.1. Data Analysis

SPSS and AMOS were used to conduct data analysis in this study. Prior to the model test, confirmatory factor analysis (CFA) was performed for the purpose of offering an estimate of the measurement model [55]. Structural equation modeling (SEM) was then used to test the theoretical-based hypotheses and model after the model was assessed and was considered adequate. In our context, the relationships among system quality, information quality, service quality, user motivation, user satisfaction, perceived net benefits, and continuance intention were examined.

### 4.2. Sample Profile and Descriptivbe Statistics

Of the 632 respondents, female respondents made up 21.2% (n = 134). Male respondents accounted for 78.8% (n = 498). The age of the respondents mostly focused on 18 to 25 years old (n = 479, 75.8%). The respondents whose ages were 26 and above accounted for 24.2% (n = 153). The respondents who had graduated from college/university (n = 506, 80.1%) and high school/vocational school (n = 94, 14.9%) were the primary categories of educational levels. The majority of the respondents revealed that they used an Android system (n = 435, 68.8%). The remaining respondents who employed an IOS system were about 30.7%. Approximately 73.3% of the respondents (n = 463) reported at least one-year’s experience of using streaming platforms.

Table 1 reports the descriptive statistics and item loading concerning the constructs of this study. On average, the respondents’ responses were positive. The mean scores of the constructs were all over 4.93 out of 7. The item loadings range from 0.77 to 0.91 on their proposed constructs. Such values exceed the minimum levels of 0.50 suggested by Hair et al. [56].

### 4.3. Measurement Model

Confirmatory factor analysis (CFA) was performed to examine whether the proposed constructs were reliable and valid. The CFA results are shown in Table 1 and Table 2. Item loadings were all ranged from 0.77 to 0.91 and exceeded the suggested value of 0.50. Table 2 shows the results and includes the measures of correlation, reliability coefficient, and average variance extracted (AVE). Composite reliability (CR) was performed in order to further examine each construct. The CR values ranged from 0.84 to 0.93. And exceeded the value of 0.70 recommended by Bagozzi and Yi [57]. AVE was also performed to analyze the convergent validity concerning the measures. The AVE values ranged from 0.64 to 0.82. Such values were larger than the value of 0.50 recommended by Fornell and Larcker [58]. The AVE square root values all exceeded the correlations between corresponding constructs. This indicates that discriminant validity was satisfactory.

### 4.4. Tests of Structural Model

Based on the IS Success Model, this study aimed to test the significance of the proposed hypotheses. The SEM was employed, and the structural model was assessed via χ^2^/df and fit indices (χ^2^/df = 2.200, GFI = 0.940, AGFI = 0.923, RMSEA = 0.044, CFI = 0.969, NFI = 0.944). The result, χ^2^/df = 2.200, was smaller than the standard score of 3.0 recommended by Bentler and Bonett [59]. The GFI had a score of 0.940, exceeding the score of 0.80 suggested by Hair et al. [56]. The AGFI had a score of 0.923, exceeding the score of 0.80 suggested by Scott [60]. The RMSEA had a score of 0.044, which met the score (less than 0.08) suggested by Hair et al. [56]. The CFI had a score of 0.969, exceeding the score of 0.90 suggested by Brown and Cudeck [61]. The NFI had a score of 0.944, exceeding the score of 0.90 suggested by Brown and Cudeck [61]. The goodness-of-fit indices revealed a satisfactory fit concerning the hypothesis model and the data.

Figure 4 presents the results of the hypothesis testing. The estimates of the standardized coefficients revealed that the linkages between system quality and user satisfaction (β = 0.224, *p* < 0.001), between information quality and user satisfaction (β = 0.234, *p* < 0.001), between service quality and user satisfaction (β = 0.341, *p* < 0.001), between user motivation and user satisfaction (β = 0.221, *p* < 0.001), between user satisfaction and perceived net benefits (β = 0.602, *p* < 0.001), between user satisfaction and continuance intention (β = 0.451, *p* < 0.001), and between perceived net benefits and continuance intention (β = 0.319, *p* < 0.001) were all positive and significant. Hypotheses 1, 2, 3, 4, 5, 6, and 7 were, therefore, supported. The findings for these hypotheses revealed that system quality, information quality, service quality, and user motivation all seemed to impact user satisfaction. User satisfaction would impact perceived net benefits and continuance intention to use. Perceived net benefits would impact continuance intention to use. In addition, as shown in Figure 2, the estimates of the standardized coefficients pointed out that the effect of service quality on user satisfaction was greater than system quality, information quality, and user motivation. The effect of user satisfaction on perceived net benefits was greater than continuance intention to use.

The moderating variable (i.e., emotional involvement) was introduced to test H8 and H9. The guidelines suggested by Hayes and Matthes [62] were followed to examine the moderating effect concerning the research model. A positive and significant moderating effect of emotional involvement was found between user satisfaction and perceived net benefits (β = 0.224, *p* < 0.001). Specifically, the relationship between user satisfaction and perceived net benefits was moderated by emotional involvement. The moderating effect between user satisfaction and continuance intention to use was not significant (β = 0.09, *p* > 0.05). Therefore, H8 was supported, whereas H9 was not supported.

## 5. Discussion, Implications, and Limitations

### 5.1. Discussion

The paper endeavored to develop and examine a theoretical explanation concerning the formation of client users’ satisfaction with, perceived net benefits of, and continuance intention to use, a particular streaming platform. The research team extended the IS Success Model by including the user motivation construct in the first order. This was because users may consider streaming platforms as a medium in providing programs for entertaining, improving gaming skills, or interacting with other users. Indeed, Lessel et al. [63] found that streaming programs such as live gaming competition and chat room communication were attractive to a wide array of audiences. Overall, the results of this study verified that the first-order constructs from the IS Success Model were likely to be the major reasons for satisfying streaming platform users as well as the possibility of this satisfaction contributing to the perception of gaining net benefits and the intention to use the streaming platform in a continuous fashion. In summary, the higher the system quality, information quality, service quality, and user motivation perceived by users, the more satisfied they were with the streaming platform used. Subsequently, streaming platform users were more likely to perceive that they have gained knowledge or experience from the platform, and they are more likely to use that particular streaming platform in a continuous manner.

Additionally, by including the moderating variable in the research model, the moderating effect of emotional involvement between user satisfaction and perceived net benefits was confirmed. The positive impact of user satisfaction on perceived net benefits was stronger for those streaming platform users who were highly emotionally involved. Therefore, emotional involvement could serve as an indicator that strengthened the relationship between user satisfaction and perceived net benefits in this study. If users spent long hours on streaming platform programs and those programs came with affective music, moving pictures, and communication channels, there was a higher level of emotional involvement.

### 5.2. Implications for Practice

The major findings of the study have significant managerial and strategy implications for the management teams of streaming platforms to use and apply the satisfaction relationships discovered in this study. The findings revealed that system quality, information quality, service quality, and user motivation all significantly affect user satisfaction. Therefore, access, user friendliness, and the interactivity of platform systems should be noticed, developed, and emphasized by the management teams of streaming platforms. Accuracy, completeness, and understandability of information provided by streaming platforms are the essence of information quality for client users. The characteristics of service quality for streaming platforms include responsiveness, reliability, and interpersonal reactivity. Paying attention to the factors investigated in this study, and their relationship to each other, would be both intrinsically and extrinsically important to streaming platform management teams. Users were motivated to use a streaming platform if the programs provided enabled users to interact with other streamers, improve their gaming skills, and have fun. Thus, partnering with programs that focus on the functions of interactivity, skill learning, and fun in an online environment would be a trend for future development.

Similar to the results reported by researchers, the findings also pointed out that user satisfaction with a streaming platform significantly affects their perceived net benefits [27,38,39] and continuance intention to use the platform [42,43]. These relationships imply that user satisfaction serves as the cornerstone of connecting user perceptions and intentions. Thus, for the purpose of better management, providing quality services which consider and address the antecedent variables of the updated IS Success Model which motivates users are the strategies which enable users to accumulate benefits and, thereby, to develop intentions to use the platform in a continuous fashion. The moderating effect of emotional involvement addresses the fact that streaming platforms need to maintain good relationships with the highly involved users in order to increase market share or numbers of client users.

### 5.3. Implication for Theory

The updated IS Success Model is a parsimonious presentation that attempts to provide a comprehensive understanding of how IS can be successfully used by users. The model employs six dimensions to describe and explain the relationships which are commonly assessed by researchers. This study extended the model by adding user motivation as the first-order construct and modifying the intention to use construct to be one of the third-order constructs (i.e., continuance intention to use). The result of this study revealed that user motivation and user satisfaction were positively and significantly correlated (β = 0.22, *p* < 0.001). This indicates that a user with higher levels of motivation was more likely to be satisfied with the programs provided by a streaming platform.

In addition, the construct ‘intention to use’ was renamed (i.e., continuance intention to use) and was modified as a third-order construct. This modification was based on the fact that the survey respondents of the study were current game streaming users. If applying the updated IS Success Model, the ’current user’ situation would hardly fit the statement addressed by Petter et al. [8], “increased user satisfaction will lead to a higher intention to use, which will subsequently affect use” (p. 238). Therefore, a modification was made and the construct ‘continuance intention to use’ was developed in the third order of the research model. These modifications highlighted the flexibility of the updated IS Success Model which can be applied in different IS settings [64,65] and fields of study [66,67].

The addition of emotional involvement as the moderating variable also extended the updated IS Success Model by implying that engaged client users could immerse in a mediated environment where emotionally touching scenes and music were presented to attract audiences. At the same time, online gaming programs and chat rooms might play a major role of enabling players to engage in communications and to build positive social relationships. As a result, the positive impact of user satisfaction on perceived net benefits was stronger for those highly emotionally involved streaming platform users.

## 6. Conclusions and Suggestions for Future Research

This study endeavored to test the appropriateness of the Information Systems Success Model in explaining streaming platform users’ satisfaction and the effect on perceived net benefits and continuance intention. The results from the updated IS Success Model (i.e., system quality, information quality, and service quality) were in line with the results of prior studies [16,65,68]. The result of extended construct (i.e., user motivation) was also in line with studies conducted by previous researchers [34,35,69]. Similar to results reported by Wang and Liao [65], user satisfaction had a positive and significant effect on perceived net benefits. User satisfaction also had a positive and significant effect on continuance intention to use a streaming platform. In addition, perceived net benefits positively and significantly related to continuance intention to use a particular streaming platform.

The study employed the updated IS Success Model as the cornerstone of developing a reference model for a better understanding of streaming platform users’ perceptual reasoning processes. The proposed model is able to provide the management teams of streaming platforms with a theoretical framework developed from consumers’ self-reported perspectives. The validity and reliability of the model can be further examined by inviting respondents from different groups or countries. For more insights, comparative studies can be conducted between developed and developing countries due to the differences in the numbers of users as well as network coverages. The study also employed emotional involvement as a moderating variable. Other moderating variables, like the level of gaming skills, or self-efficacy, can be used to yield additional insights into the IS Success Model as well as the perceptions of game streaming platform users. Finally, other theoretical frameworks such as the Unified Theory of Acceptance and Use of Technology and the Theory of Planned Behavior can be employed and tested for the purpose of developing more comprehensive models for future research and discovery.

The limitations of the study consist of sampling strategy and sample size. Purposeful sampling was employed in this study. The results of the study cannot be generalized. If a complete population list is available and random sampling can be used as the sampling strategy, more thorough findings can be obtained. Additionally, the majority of the respondents were young adults (i.e., 18–25 years old, 75.8%) and male respondents (78.8%). Having more equal representation of age and gender groups may present different findings.

## Figures and Tables

**Figure 1 behavsci-12-00124-f001:**
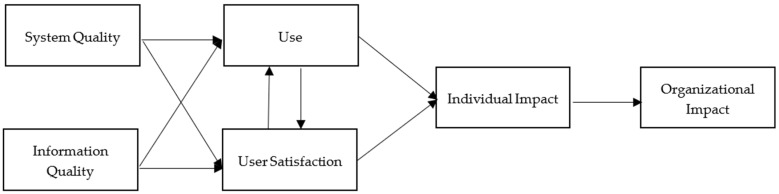
The Original IS Success Model [11].

**Figure 2 behavsci-12-00124-f002:**
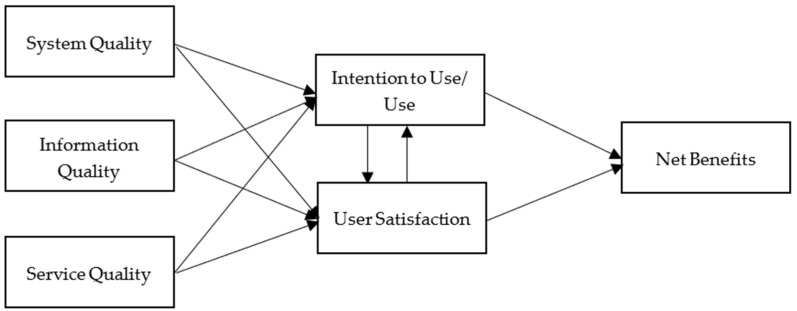
The Updated IS Success Model [11].

**Figure 3 behavsci-12-00124-f003:**
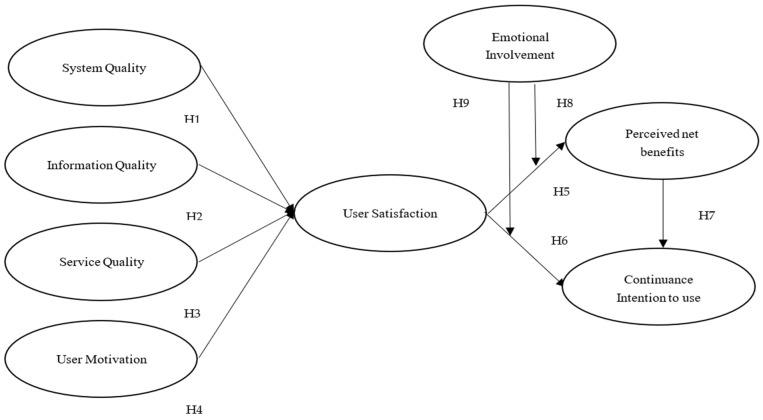
Research framework.

**Figure 4 behavsci-12-00124-f004:**
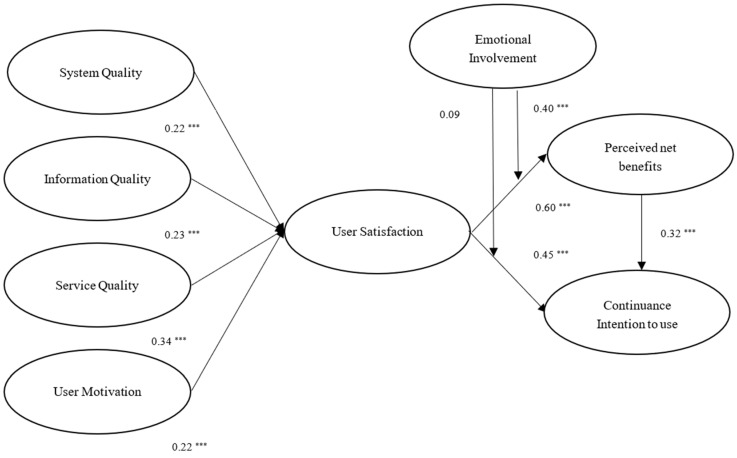
Results of hypothesis analysis. *** *p* < 0.001.

**Table 1 behavsci-12-00124-t001:** Descriptive statistics and item loading.

Variable	Mean	SD	Loading
**System quality**			
1. I think the connection to the streaming platform is stable.	5.92	0.92	0.88
2. I think the streaming platform is user friendly.	5.71	0.97	0.90
3. I think the streaming platform provides instant interactions smoothly.	5.44	1.06	0.87
**Information quality**			
1. I think the streaming platform provides accurate information for game streaming users.	5.46	1.19	0.84
2. I think the streaming platform contains complete information required for game streaming users.	5.59	1.10	0.81
3. I think the information provided by the streaming platform is easy to understand for game streaming users.	5.37	1.22	0.87
**Service quality**			
1. I think the notification service of the streaming platform enables game streaming users to watch games in real-time.	5.08	1.25	0.78
2. I think the service of the streaming platform is reliable.	4.72	1.15	0.80
3. I think the live interpersonal service provided by the streaming platform makes me feel a sense of participation.	4.98	1.11	0.88
**User Motivation**			
1. The streaming platform enables me to interact with other streamers.	5.21	1.16	0.90
2. The streaming platform enables me to improve their gaming skills.	5.23	1.10	0.90
3. The streaming platform enables me to have fun.	5.13	1.12	0.88
**User satisfaction**			
1. I am satisfied with the stability of the system provided by the streaming platform.	5.36	1.07	0.83
2. I am satisfied with various entertainment functions provided by the streaming platform.	5.21	1.10	0.84
3. I am satisfied with the information exchange function provided by the streaming platform.	5.24	1.05	0.87
4. I am satisfied with the gaming programs provided by the streaming platform.	5.29	1.04	0.87
**Perceived net benefits**			
1. I would share my gaming experiences with other members via the streaming platform.	5.06	1.29	0.91
2. Members of the streaming platform would share their ideas/opinions with me.	5.08	1.22	0.90
3. Members of the streaming platform helps improve my gaming performance.	4.85	1.32	0.90
**Continuance intention to use**			
1. I like to use the streaming platform.	5.44	0.99	0.83
2. I am willing to introduce the streaming platform to my relatives/friends.	5.12	1.21	0.77
3. I am willing to use the streaming platform in the future.	5.14	1.13	0.80

1 = Strongly disagree, 2 = Disagree, 3 = Somewhat disagree, 4 = Neutral, 5 = Somewhat agree, 6 = Agree, 7 = Strongly agree (n = 632).

**Table 2 behavsci-12-00124-t002:** Measures of correlation, composite reliability, and average variance extracted.

Construct	1	2	3	4	5	6	7	AVE
1. System quality	0.88							0.78
2. Information quality	0.58	0.84						0.71
3. Service quality	0.36	0.44	0.82					0.67
4. User’s Motivation	0.48	0.51	0.51	0.89				0.80
5. User satisfaction	0.55	0.58	0.58	0.60	0.85			0.73
6. Perceived net benefits	0.37	0.37	0.34	0.43	0.53	0.90		0.82
7. Continuance intention	0.47	0.44	0.40	0.38	0.50	0.49	0.80	0.64
Composite reliability	0.92	0.88	0.86	0.92	0.92	0.93	0.84	

Note: Diagonals represent the square root of AVEs.

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
