# Peer review of "Continuance Intention to Use and Perceived Net Benefits as Perceived by Streaming Platform Users: An Application of the Updated IS Success Model"

_behavsci, 2022, doi:10.3390/bs12050124_

Round 1

Reviewer 1 Report

This study aimed at developing and examining a theoretical model based on an updated information system to explore the relationship among users’ satisfaction with, perceived net benefits of, and continuance intention to use a particular streaming platform. Overall, this study was well-organized and meaningful for practice and theory. Below are some suggestions for quality improvement.

1. In literature review and hypothesis development section, the IS success model was introduced in 2.1. while there was not enough explanation about the updated part of this model. It is suggested to make a minor modification of the title to “Updated IS Success Model” and describe more about the updated contents that link to the current study.

2. Clarification of the reasons why the original four dimensions of LSM in Beard and Ragheb study was reduced to three in this study was needed.

3. In the method section-3.1 section, has factor analysis been conducted for the pilot test? If yes, the results could be a supportive evidence for the construct validity.

4. In 3.2, it is suggested to add the description of sampling strategy used in this study.

5. For the figure 2, all variables with rectangles should be replaced by circle shapes for the SEM.

6. There is a minor repetition in “discussions” section and “conclusions and suggestions for future research”. 

Reviewer 2 Report

My main concern about this article is the title. It is a research question and not a title, which creates an immediate barrier for the reader.

Before H1, the main study aim is missing. The authors present in detail different hypotheses, but there is no cohesive main aim that sums up the entire investigation.

Another important flaw of this article is that a cohesive and extensive reflection around the limitations is completely missing. This needs to be resolved.

Reviewer 3 Report

The paper deals with an interesting topic of using information systems quality assessment model to evaluate satisfaction with computer games streaming platforms.

The structure of the paper is appropriate, as well as scientific methodology. Relevant literature has been addressed. 

Authors are suggested to emphasize the contribution regarding the announced "Updated Information Systems Success Model".

1) First, there should be clearly stated what is the "basic" Information Systems Success Model and what is the update that they propose. There should be also correspondence between user satisfaction and the proposed new, i.e. "Updated Information Systems Success Model". It is suggested that authors provide a new section that will clearly describe the contribution. This section should be put before research methods, which are focused on questionnaire, research sample and data analysis. It will address and graphically present (similar to Figure 1): 1) existing Information Systems Success Model, 2) Updated Information Systems Success Model, 3) correspondence between user satisfaction criteria and Updated Information Systems Success Model. This section is crucial for this paper, since it is announced in the title of the paper. Without this section, it will not be possible to keep the current title of the paper.

2) Second, the conclusion should emphasize the contribution with the updated information systems success model. In conclusion, there is DeLone and McLean’s updated information systems success model mentioned as a basis for this research. 

3) Finally, if authors do decide not to change the content (since they do not propose the update, but use existing one), then the title should be changed. Instead of "updated information systems success model" (currently the readers could think that authors propose the "update") into DeLone and McLean’s updated information systems success model.
